# The Anti-Diabetic Potential of Baicalin: Evidence from Rodent Studies

**DOI:** 10.3390/ijms25010431

**Published:** 2023-12-28

**Authors:** Tomasz Szkudelski, Katarzyna Szkudelska

**Affiliations:** Department of Animal Physiology, Biochemistry and Biostructure, Poznan University of Life Sciences, Wołyńska 35, 60-637 Poznań, Poland; katarzyna.szkudelska@up.poznan.pl

**Keywords:** hyperglycemia, metabolic dysregulation, skeletal muscle, adipose tissue, liver

## Abstract

Baicalin is a biologically active flavonoid compound that benefits the organism in various pathological conditions. Rodent studies have shown that this compound effectively alleviates diabetes-related disturbances in models of type 1 and type 2 diabetes. Baicalin supplementation limited hyperglycemia and improved insulin sensitivity. The anti-diabetic effects of baicalin covered the main insulin-sensitive tissues, i.e., the skeletal muscle, the adipose tissue, and the liver. In the muscle tissue, baicalin limited lipid accumulation and improved glucose transport. Baicalin therapy was associated with diminished adipose tissue content and increased mitochondrial biogenesis. Hepatic lipid accumulation and glucose output were also decreased as a result of baicalin supplementation. The molecular mechanism of the anti-diabetic action of this compound is pleiotropic and is associated with changes in the expression/action of pivotal enzymes and signaling molecules. Baicalin positively affected, among others, the tissue insulin receptor, glucose transporter, AMP-activated protein kinase, protein kinase B, carnitine palmitoyltransferase, acetyl-CoA carboxylase, and fatty acid synthase. Moreover, this compound ameliorated diabetes-related oxidative and inflammatory stress and reduced epigenetic modifications. Importantly, baicalin supplementation at the effective doses did not induce any side effects. Results of rodent studies imply that baicalin may be tested as an anti-diabetic agent in humans.

## 1. Introduction

Baicalin is a plant-derived biologically active compound. It is present in high amounts in *Scutellaria baicalensis* [1]. Plants abundant in baicalin are widely applied in traditional medicine for treating inflammation-related diseases, dysentery, hypertension, hemorrhage, insomnia, diabetes and its complications, and others [2,3,4,5]. Apart from therapies based on plants rich in baicalin, the pure compound is recommended in humans as a dietary health-promoting supplement [2,6]. Baicalin (7-glucuronic acid, 5,6-dihydroxy-flavone; Figure 1) is a polar glycosidic flavone. Its water solubility is relatively low and reaches about 70 mg/L. Due to its chemical properties, the absorption of baicalin from the digestive tract is quite small. It is estimated that only a few percent of the ingested compound is absorbed. However, a large part of baicalin undergoes conversion to its aglycone form, baicalein, the absorption of which is much higher. It was also shown that, after baicalin ingestion, this compound appears in the blood and reaches numerous tissues [7,8]. Despite relatively low absorption, baicalin supplementation was effective in various pathological conditions. This points out that baicalin has a large therapeutic potential. Moreover, interest in baicalin as a natural health-promoting agent is still increasing [2,6].

Baicalin consumption is associated with numerous health advantages, including the prevention and treatment of various diseases. Results of animal studies have proven its efficacy in alleviating different pathological conditions. It has a therapeutic role in inflammatory bowel disease [1,9], rheumatoid arthritis [8], pulmonary hypertension [10], metabolic syndrome [3,11], cardiovascular diseases [12], and also some kinds of cancer [13]. Although the reach of brain baicalin is not fully proven, this compound effectively alleviates neurodegenerative disorders [7,14]. Moreover, baicalin was shown to have hepatoprotective effects and mitigate viruses such as hepatitis, fatty liver diseases, hepatocellular carcinoma, and cholestatic liver injury. It is also well established that this compound protects the liver against injury induced by various xenobiotics [6,15]. Baicalin exerts anti-inflammatory and immunomodulatory action and thereby alleviates inflammation-related disorders. Baicalin was also shown to be effective in limiting oxidative stress [12]. Both inflammation and oxidative stress are largely related to each other and are a background of different diseases. The anti-inflammatory and antioxidant activity of baicalin is therefore important for its therapeutic efficacy [4,12].

The molecular background of baicalin action at the cellular level is associated with changes covering pivotal enzymes and regulatory molecules. Its supplementation may affect sterol regulatory element-binding protein 1 (SREBP-1), peroxisome proliferator-activated receptors (PPARs), and the mammalian target of rapamycin (mTOR). Influencing SREBP-1, PPARs, and mTOR, baicalin regulates gene expression at both transcriptional and translational levels [10,13,15]. This compound may also influence two important enzymes, i.e., mitogen-activated protein kinase (MAPK) and silent information regulator 1 (SIRT1). MAPK is involved in the cellular response to different stress conditions, such as osmotic stress, heat shock, and inflammation [16]. SIRT1 is an NAD^+^-dependent histone deacetylase involved in regulating mitochondrial biogenesis, inflammation, stress resistance, apoptosis, and others [17]. Baicalin has also been implicated in the regulation of cellular metabolism. This compound was shown to affect AMP-activated protein kinase (AMPK) and acetyl-CoA carboxylase (ACC) [10,13,15]. AMPK is a cellular energy sensor activated under hypoxia, ischemia, glucose deficiency, caloric restriction, and physical activity. Activation of AMPK is associated with alterations in metabolic pathways related to energy formation and usage. ACC is one of the pivotal AMPK targets and catalyzes the generation of malonyl-CoA from acetyl-CoA. AMPK phosphorylates ACC, which in this way makes it inactive, decreasing malonyl-CoA synthesis and promoting β-oxidation. It was also demonstrated that baicalin can affect signaling molecules involved in hormone action, such as protein kinase B (PKB/Akt), phosphoinositide 3-kinase (PI3K), and others. This is associated with changes in hormonal signaling [10,13,15].

It is well established that baicalin exerts pleiotropic effects, including anti-diabetic. Diabetes in humans is associated with grave health complications, including diabetic foot, decreased vision, infarction, renal failure, and others, the occurrence of which may be effectively alleviated or delayed [18,19]. It was shown that baicalin and other compounds present in high amounts in *Scutellaria baicalensis*, such as baicalein, wagonoside, and wagonin, can limit diabetic nephropathy, retinopathy, cardiovascular disease, micro- and macro-angiopathy, neuropathy, and others. These effects were shown in various animal models of diabetes and also in vitro [5,6]. Thus, the mitigation of diabetes complications by compounds found in *Scutellaria baicalensis* is a valuable property and encourages clinical tests. The present article provides evidence from rodent studies that baicalin is also effective in alleviating diabetes-related hormonal and metabolic disturbances.

## 2. Diabetes in Humans

Diabetes is a grave metabolic disease affecting as many as 10% of adults worldwide. Due to its high prevalence, the disease is a serious global problem. The number of people with diabetes is observed to be still increasing. Type 1 and type 2 diabetes are the most frequent.

Type 1 diabetes covers about 10% of all cases, while type 2 diabetes is more frequent and accounts for nearly 90% of cases [18,19]. The etiology, symptoms, and ways of treatment for type 1 and 2 diabetes markedly differ. In humans with type 1 diabetes, pancreatic β-cells undergo massive destruction. This process occurs quickly and usually has an auto-immunological background. Since pancreatic β-cells are the only source of insulin, their loss causes a lack of this hormone. Insulin is the only hormone that effectively reduces blood glucose levels. Both appropriate blood insulin concentration and proper action on target cells are needed for the blood-glucose-lowering effect. As a result of insulin deficiency, blood glucose levels are elevated (hyperglycemia) and markedly exceed the physiological range. Symptoms of type 1 diabetes usually appear in a relatively short-term period. Insulin deficiency and the resulting hyperglycemia are associated with polyuria and polydipsia. The proper functioning of the nervous system is known to be fully dependent on an appropriate glucose supply. Both glucose deficiency and excess glucose are extremely detrimental to neurons. In the case of considerable hyperglycemia and the increased glucose supply, neurons undergo damage, and diabetic coma develops. Insulin deficiency and the resulting elevated blood glucose levels lead to micro- and macro-angiopathies, retinopathies, cardiopathy, nephropathies, and other grave disorders, including necrotic changes. This is associated with a progressive failure of many organs. The damage to pancreatic β-cells causes patients to be continuously treated with exogenous insulin. However, proper insulin therapy reduces blood glucose levels to the physiological range and minimizes diabetic complications [18,19,20,21,22,23].

In type 2 diabetes, both insulin secretion and action are impaired. Insulin resistance, a pivotal hallmark of type 2 diabetes, means that the action of insulin is insufficient. Type 2 diabetes develops slowly. In humans with this type of diabetes, blood glucose levels are initially normal or only moderately elevated due to the compensatory hypersecretion of insulin [18,19]. However, sustained excessive insulin secretion may induce “exhaustion” of β-cells [24]. After many years of such a condition, in patients with type 2 diabetes, type 1 diabetes develops. In this case, patients must be treated with the exogenous hormone [18,19,22]. Type 2 diabetes is often associated with low physical activity, excessive energy intake in a diet, and the resulting overweight or obesity [18,19,20,24,25,26]. However, in some populations, type 2 diabetes develops despite normal body mass [26]. Irrespective of body mass, inflammatory processes and accompanying oxidative stress have been largely implicated in the pathogenesis and progression of type 2 diabetes. In the case of overweight and obese humans, excessive adipose tissue is associated with the accumulation of macrophages. These cells release proinflammatory cytokines, thereby causing chronic low-grade inflammation and inflammatory stress. Therapies for people with type 2 diabetes include various classes of drugs, such as insulin sensitizers, insulin secretagogues, incretin analogs, biguanides, α-glucosidase inhibitors, dipeptidyl peptidase-4 inhibitors, and intestinal lipase inhibitors [18,19,20]. Although anti-diabetic drugs are very effective, their prolonged use is associated with side effects, such as gastrointestinal problems, kidney complications, and others [27,28]. Therefore, in humans with obesity-related diabetes, increased physical activity, dietary intervention, and a reduction in body mass are recommended to support the action of drugs and minimize their side effects [20,25,29]. Moreover, various natural, plant-derived compounds are constantly being tested in animal studies to assess their anti-diabetic potential [30,31,32]. Many of them have been proven to be effective in alleviating diabetes [5,33].

## 3. Anti-Diabetic Effects of Baicalin

### 3.1. Anti-Hyperglycemic Action of Baicalin

Under physiological conditions, blood glucose levels need to be maintained in a very narrow range. Insulin is the sole hormone exerting clear-cut anti-hyperglycemic action. Its secretion is precisely regulated to avoid hyper- or hypoglycemia. Glucose is the main physiological agent stimulating insulin secretion. After a meal, blood glucose levels are elevated, which is associated with increased glucose transport into pancreatic β-cells. This is followed by augmented oxidative glycolysis, hyperpolarization of the inner mitochondrial membrane, a rise in ATP formation and ATP/ADP ratio, the closure of ATP-dependent K^+^ channels, plasma membrane depolarization, and the opening of voltage-dependent Ca^2+^ channels. This sequence of events triggers insulin exocytosis. Along with glucose, the gut-derived incretin hormones, islet-derived adenosine, and the nervous system (secreting adrenaline and acetylcholine) have a relevant modulatory function on insulin secretion [34,35,36,37,38,39]. However, in type 1 diabetes, the insulin-secretory function of β-cells markedly deteriorates, leading to insulin deficiency and the resulting hyperglycemia. Sustained hyperglycemia is well known to largely contribute to the progressive failure of cells and organs, including insulin-secreting cells, and is one of the major causative factors of diabetic complications. On the other hand, maintaining blood glucose levels in the physiological range (euglycemia) limits the progression of diabetes and associated complications [18,19,39]. Such action is well documented for numerous natural, plant-derived compounds [33,40,41], including baicalin. The blood-glucose-lowering properties of baicalin have been shown in studies using various animal models of diabetes. This compound effectively mitigated hyperglycemia in rats [42,43,44] and mice [45,46] with diabetes induced by streptozotocin (STZ) administration. STZ selectively destroys pancreatic β-cells, thereby markedly reducing insulin content in pancreatic islets. Initially, STZ induces DNA damage through alkylation. This is followed by an exaggerated activation of poly(ADP-ribose) polymerase-1 (PARP-1) and the resulting depletion of NAD^+^ and ATP. STZ evokes diabetes with symptoms similar to type 1 diabetes in humans. This compound induces considerable blood insulin deficiency and the resulting hyperglycemia [47,48]. Sustained hyperglycemia leads to an increased generation of reactive oxygen species (ROS), which additionally impairs β-cell structure and endocrine functions [23]. Compared with other kinds of cells, pancreatic β-cells are known to be very susceptible to the action of ROS. This is due to a weak antioxidant defense system [49,50]. Importantly, it was shown that in rats with STZ-induced diabetes, baicalin therapy effectively increased pancreatic insulin content and limited pancreatic lipid peroxide content (detrimental to cell end products of ROS action) [51]. Baicalin also prevented the loss of integrity of the inner mitochondrial membrane in β-cells, which is important for proper glucose-induced insulin secretion [51]. Moreover, baicalin reduced hyperglycemia and relieved pancreatic islet damage in rats with diabetes induced by the administration of STZ and nicotinamide (NA) [52]. NA inhibits PARP-1 and provides NAD^+^, alleviating the cytotoxic action of STZ [48,53]. This proves that in animals with STZ-induced diabetes, baicalin can effectively improve the structure and endocrine function of the insulin-secreting cells. Given that insulin is the sole hormone effectively reducing blood glucose levels, and in STZ-induced diabetes, insulin deficiency is not accompanied by insulin resistance, the anti-hyperglycemic effect of baicalin is due to an elevation in blood insulin levels [46,51]. These results show that baicalin therapy is associated with a protective action on pancreatic β-cells. This is an important property of baicalin since the deterioration of β-cell function develops not only in type 1 diabetes but also in type 2 diabetes.

### 3.2. Effects of Baicalin on Insulin Resistance

Insulin action on target cells is a complex process involving many intracellular signaling molecules. The first step of insulin action, preceding cellular changes, is hormone binding to the transmembrane receptor. This triggers a sequence of events involving the autophosphorylation of the receptor and the phosphorylation of intracellular substrates. It is known that many steps of the insulin signaling pathway may be disturbed, leading to insulin resistance [20,54,55]. Insulin resistance is one of the main hallmarks of type 2 diabetes in humans. At the initial stages of the disease, the concentration of blood insulin is elevated due to compensatory hypersecretion of the hormone by β-cells. However, despite hyperinsulinemia, the action of insulin at the cellular level is impaired. Studies on animal models with insulin resistance provided evidence that baicalin effectively improves insulin sensitivity and thereby mitigates hyperglycemia. It was shown that baicalin treatment reduces blood glucose levels in rats fed a high-fat diet (HFD) [56], fed an HFD with a high-sucrose diet (HSD), and treated with STZ [57]. Similar blood glucose-lowering action was revealed in mice on an HFD [58,59,60,61], and also in Goto-Kakizaki (GK) rats [62]. GK rats are non-obese and establish a model of type 2 diabetes determined genetically [63]. Feeding rodents an HFD alone or an HFD with an HSD is associated with insulin resistance. Impaired insulin action also develops spontaneously in GK rats [63]. Baicalin therapy was shown to alleviate insulin resistance in mice fed an HFD [59,60,61,64], in GK rats [62], and also in db/db mice [65]. The db/db mice are obese and have symptoms of type 2 diabetes due to leptin gene mutation and the resulting leptin deficiency. Leptin is a hormone secreted predominantly from adipocytes, has anorectic action, and its deficiency induces obesity and insulin resistance [66].

In rodent models of insulin resistance, blood insulin levels may be increased (hyperinsulinemia) or decreased (hypoinsulinemia), depending on the experimental model. Both changes are abnormal since, in each case, they are associated with insufficient insulin action. In mice [59,60,64] and rats [56] with hyperinsulinemia induced by feeding an HFD, baicalin supplementation reduced blood insulin levels. A similar effect on blood insulin concentrations following baicalin therapy was found in GK rats [62]. Given that continued hyperinsulinemia is associated with insulin resistance, a decrease in blood insulin levels by baicalin indicates improved action of this hormone on target cells. It is well established that continued overstimulation of insulin secretion largely contributes to a progressive failure of β-cells. Moreover, clinical studies imply that the temporary limitation of exaggerated insulin secretion (“β-cell rest”) in humans with type 2 diabetes delays β-cell dysfunction [67]. Apart from overstimulating insulin secretion, another causative factor largely contributing to the progressive failure of β-cells in type 2 diabetes is sustained hyperglycemia [18,19]. Therefore, the anti-hyperinsulinemic and anti-hyperglycemic effects of baicalin indicate that this compound effectively limits β-cell degradation. This is a relevant effect since it indicates that baicalin is capable of delaying the progression of type 2 diabetes.

These results show that baicalin supplementation limits hyperglycemia resulting from both insulin deficiency and insulin resistance. Moreover, baicalin was demonstrated to be effective in alleviating insulin resistance induced by diet and determined genetically (Table 1). The proper blood insulin concentrations and action of this hormone at the cellular level are needed to effectively reduce hyperglycemia. The blood-glucose-lowering effects of insulin require action covering three metabolically active tissues: the skeletal muscle, the liver, and the fat tissue.

### 3.3. Effects of Baicalin on the Skeletal Muscle

The skeletal muscle is the main tissue responsible for the blood-glucose-lowering action of insulin. Under physiological conditions, this tissue is responsible for a major part of the insulin-stimulated blood-glucose clearance. However, skeletal muscle insulin resistance largely contributes to hyperglycemia. Therefore, in diabetic states, improved skeletal muscle insulin action is crucial for whole-body glucose homeostasis [20]. It was demonstrated that baicalin therapy effectively alleviates insulin resistance in the skeletal muscle. This important effect was largely related to the effects of baicalin on blood lipid contents in diabetic rodents. Baicalin treatment reduced blood levels of non-esterified fatty acids (NEFAs) in insulin-resistant mice [59,64], rats on an HFD [56], and rats with diabetes induced by feeding an HFD/HSD and administration of STZ [57]. Moreover, baicalin supplementation diminished concentrations of blood triglycerides (TGs) in HFD-fed rats [56], in insulin-resistant GK rats [62], and in obese mice [59,68]. The effects of baicalin on blood lipids and the resulting intramuscular changes are the relevant elements in the mechanism of its anti-diabetic action. The role of increased blood NEFA and TG content in the pathogenesis and progression of insulin resistance is well established. Elevated concentrations of blood NEFAs and TGs, especially those containing long-chain saturated fatty acids, induce an exaggerated supply of these compounds to the muscle tissue. This leads to the increased intracellular formation of ROS, fatty-acyl-CoA, ceramide, and diacylglycerol (DAG). These detrimental changes induce protein kinase C (PKC0) activation. PKC0 catalyzes the phosphorylation of the main insulin-receptor substrates (IRS1 and IRS2) at Ser/Thr residues, which restrains insulin signaling. Under these conditions, the pathway insulin receptor—IRS1,2—phosphatidylinositol 3-kinase (PI3K)—PKA—GLUT4 is downregulated, and insulin resistance develops [69,70,71,72,73]. Moreover, in the pathogenesis of skeletal muscle insulin resistance, the AMPK-ACC pathway is also involved. The latter enzyme catalyzes the conversion of acetyl-CoA to malonyl-CoA. Malonyl-CoA is thought to be a signaling molecule regulating fatty acid metabolism [71]. It was shown that baicalin effectively reduces skeletal muscle lipid accumulation in mice fed an HFD. Baicalin was demonstrated to alleviate HFD-induced reduced phosphorylation of AMPKα (at Thr172) and the resulting phosphorylation (at Ser79) and inhibition of ACC [59]. The baicalin-induced drop of intracellular malonyl-CoA promotes the mitochondrial degradation of fatty acids and simultaneously decreases their synthesis. These positive alterations effectively limit intramyocellular lipid content, improving insulin sensitivity [70,71,72,73,74].

Baicalin therapy was shown to be associated with beneficial changes covering various proteins related to insulin signaling and action in the skeletal muscle. Supplementation of this compound to mice attenuated HFD-induced inhibition of phosphorylation of both skeletal muscle protein kinase B (PKB/Akt, at Thr308) and glycogen synthase kinase 3 beta (GSK-3β, at Ser9) [59]. Moreover, baicalin elevated the expression levels of peroxisome proliferator-activated receptor-gamma coactivator-1 α (PGC-1α), p38 mitogen-activated protein kinase (p38MAPK), and protein p-AS160. Importantly, the expression of glucose transporter GLUT4 in the muscle tissue of obese mice was also up-regulated by baicalin therapy [61,74]. These effects indicate improved insulin action, especially insulin-stimulated glucose transport (via GLUT4) into the muscle cells. This effect is of great relevance since, without insulin-induced GLUT4 activation, glucose transport into myocytes is insufficient to ensure appropriate blood-glucose clearance.

Galanin has been shown to be implicated in the positive changes induced by baicalin in the skeletal muscle of obese mice. Under physiological conditions, galanin regulates cellular glucose metabolism and exerts anti-hyperglycemic action after binding to its receptor. The blood-glucose-lowering effect of this hormone is due to a rise in glucose uptake by the skeletal muscle, the liver, and the adipose tissue. However, sustained, exaggerated blood galanin concentrations are associated with galanin resistance, impaired glucose tolerance, and insulin resistance. In obese mice, blood galanin levels were excessively elevated; however, baicalin treatment reduced the content of this hormone [61]. The importance of galanin in baicalin action was confirmed by findings showing that the positive effects on the skeletal muscle were abrogated by the galanin receptor antagonist (M871) [61]. The involvement of galanin in baicalin action is a vital finding since the positive correlation between blood galanin levels and insulin resistance was revealed not only in animal models but also in patients with type 2 diabetes [3,61]. The anti-diabetic effects of baicalin on the skeletal muscle are shown in Table 2. The suggested mechanisms of baicalin action in skeletal muscle cells are presented in Figure 2.

### 3.4. Effects of Baicalin on the Adipose Tissue

Adipose tissue has many relevant functions in the organism, including energy storage and adipokine secretion. Energy is stored in the form of TGs. These compounds are formed after a meal (lipogenesis) or decomposed (lipolysis) in the post-absorptive state. Under physiological conditions, there is a balance between lipogenesis and lipolysis to avoid fat excess or deficit in the body. In the post-absorptive state, TGs stored in adipocytes are decomposed into glycerol and NEFAs. Both glycerol and NEFAs are released in the appropriate amounts into the blood, reach other tissues, and are metabolized. However, in the case of overweight or obesity, glycerol and NEFA release from adipocytes is excessive. This is associated with impaired glucose tolerance and insulin resistance [20,50,69,75,76]. It is also well established that in obesity, adipocyte size is increased [77]. Baicalin supplementation to rodents with insulin resistance was shown to be associated with reduced body fat content. This beneficial effect, demonstrated in rats [56] and mice [68,78] fed an HFD, largely contributes to better insulin sensitivity.

Obesity is associated with pathological adipose tissue remodeling. Among various obesity-related changes, immune cell infiltration has relevant implications. Macrophages, which accumulate within adipose tissue, secrete inflammatory cytokines. This evokes chronic low-grade inflammation and inflammatory stress. Inflammatory stress is one of the relevant factors contributing to impaired insulin action, insulin resistance, and type 2 diabetes [20,79,80,81,82]. Baicalin therapy was shown to alleviate these unfavorable changes effectively. It was demonstrated that in mice on an HFD, baicalin decreased adipose tissue content, reduced tissue macrophage content, and down-regulated expression of the pro-inflammatory cytokine tumor necrosis factor-α (TNF-α) [64]. This indicates that baicalin has an anti-inflammatory action in the adipose tissue of obese mice. The anti-inflammatory effects of baicalin were shown in many other pathological conditions [1,9].

It is known that some beneficial effects of baicalin on adipose tissue are mediated by AMPK. In the adipose tissue of mice with insulin resistance induced by feeding an HFD, baicalin increased mitochondrial biogenesis in an AMPK-dependent manner [83].

Apart from energy storage, adipocytes also release adipokines, hormones with important regulatory roles. Secreting adipokines, fat tissue is involved in regulating feeding behavior, energy expenditure, metabolism, and many more. However, concentrations of adipokines in the blood are abnormal in diabetes, which impairs their physiological functions. Baicalin supplementation in animals with hypoleptinemia was shown to increase blood leptin levels. This effect was found in insulin-resistant GK rats [62] and in rats with diabetes induced by STZ [42,51]. Fat cells are the main source of circulating leptin [84,85]. This hormone is well established as a relevant adipocyte-derived signal regulating energy expenditure, body weight gain, glucose utilization, lipid metabolism, and insulin sensitivity [86]. Both prolonged hyperleptinemia and leptin deficiency have grave hormonal and metabolic implications. Leptin and insulin interact with each other to regulate glucose homeostasis. Under physiological conditions, leptin improves hepatic insulin sensitivity, suppresses glucose production, and increases peripheral glucose uptake, whereas hyperleptinemia or leptin deficiency is usually associated with hyperglycemia and insulin resistance [85,87]. Thus, the increase in blood leptin content shown following baicalin therapy contributes to alleviating diabetes-related pathological changes [42,51,62]. Figure 3 and Table 3 summarize the effects of baicalin on adipose tissue.

### 3.5. Effects of Baicalin on the Liver

Baicalin was shown to have hepatoprotective effects in various pathological conditions [6,15]. The liver is a multifunctional organ and is involved in many vital processes, including glucose homeostasis. After a meal, the liver stores the excess glucose absorbed from the digestive tract as glycogen. In the post-absorptive state, the liver releases into the blood glucose derived from glycogen or gluconeogenesis. The balance between hepatic glucose accumulation and release is pivotal for maintaining blood glucose levels in the physiological range. The processes related to glucose accumulation and release are largely regulated by insulin and are disturbed in insulin-resistant states. Thus, proper insulin sensitivity in the liver is of high relevance for whole-body glucose homeostasis [88,89]. One of the relevant physiological factors implicated in the regulation of gluconeogenesis is blood NEFAs. Under physiological conditions, after a meal, insulin downregulates lipolysis in adipocytes and blood NEFAs decrease, which is a signal for the inhibition of hepatic gluconeogenesis, while in the post-absorptive state, blood NEFAs increase and gluconeogenesis rises. However, in diabetes-related insulin resistance, the regulatory function of NEFAs is disturbed. In this case, insulin does not suppress adipocyte lipolysis, blood NEFAs are permanently elevated, and gluconeogenesis is excessive [20]. Studies using animal models of diabetes have shown that baicalin therapy is associated with alleviating insulin resistance in the liver. Baicalin-induced improvement in insulin sensitivity covers various mechanisms. One of the relevant is diminished liver lipid accumulation. The link between augmented hepatic fat content and insulin insensitivity in diabetes is well established. These changes are similar in the liver and the skeletal muscle. Exaggerated NEFA supply induces metabolic changes in the liver. In this case, NEFAs are converted to fatty acyl-CoA, which evokes the augmented generation of lysophosphatidic acid, DAG, and TGs (lipogenesis). Increased hepatic DAG levels induce the translocation of PKC to the plasma membrane and inhibit insulin receptor tyrosine kinase activity. The inhibition is due to the autophosphorylation of the insulin receptor at Thr1160. This is followed by the inactivation of insulin receptor substrate 2 (IRS-2) and other signaling proteins, such as phosphatidylinositol 3-kinase (PI3K) and Akt [73,76]. Baicalin effectively limited liver lipid accumulation in mice [58,90] and rats [56,91] with insulin resistance induced by feeding an HFD. This effect is due to changes in the expression and activity of hepatic enzymes, both indirectly and directly related to lipid metabolism. In rats [56] and mice [59] subjected to HFD, baicalin therapy increased hepatic phosphorylation of AMPKα (at Thr172) and ACC (at Ser79). The latter enzyme is one of the pivotal AMPK targets, especially in the context of intracellular regulation of lipid metabolism. The active ACC catalyzes the conversion of acetyl-CoA to malonyl-CoA. Reduced formation of malonyl-CoA promotes mitochondrial degradation of FAs (fatty acids) and thereby limits intracellular lipid content. Phosphorylation of AMPK (pAMPK) induces phosphorylation of ACC (pACC) at the inhibitory site of the enzyme, which diminishes lipid deposition. Moreover, in HFD-fed mice, baicalin reduced the amount of hepatic acetyl-CoA, which additionally decreased malonyl-CoA generation and thereby accelerated FA β-oxidation [90].

Another relevant enzyme implicated in the regulation of lipid metabolism and affected by baicalin is carnitine palmitoyltransferase (CPT). It was shown that in mice fed an HFD, baicalin increased the activity of carnitine palmitoyltransferase 1A (CPT1A). CPT1A, the hepatic isoform of CPT, is linked to the outer mitochondrial membrane and is the rate-limiting step of β-oxidation [92]. Baicalin was proven to bind to CPT1A directly and is its allosteric activator [58]. This enzyme catalyzes an essential step in the β-oxidation of long-chain fatty acids, and its upregulation effectively restrains intracellular lipid accumulation. The relevance of CPT1A to baicalin action has been confirmed in studies on mice with CPT1A deficiency. In this experimental model, baicalin was ineffective in preventing the formation of liver fat depots [58]. The importance of CPT1 deficiency in the pathogenesis of insulin resistance is also suggested in humans [93]. Moreover, the dysfunction of the CPT system and the resulting abnormal lipid metabolism are associated with various other diseases [92].

Another vital enzyme involved in intracellular lipid metabolism and affected by baicalin is fatty acid synthase (FAS). Baicalin supplementation in rats fed an HFD down-regulated hepatic expression of FAS. This effect restrains the formation of fatty acids, thereby additionally reducing intracellular lipid content [56]. This data indicates that in animal models of diabetes, baicalin limits hepatic lipid content via two main mechanisms, i.e., increased FA degradation and reduced FA formation.

Apart from alleviating lipid accumulation and resulting in better insulin sensitivity, it was shown that baicalin therapy positively affects parameters related to hepatic glucose transport and metabolism. In HFD-fed mice, baicalin upregulated the expression of the glucose transporters GLUT1 and GLUT4 [60]. This indicates improved glucose transport into hepatocytes and the resulting better blood glucose clearance following baicalin therapy. Moreover, baicalin reduced the expression of key enzymes catalyzing glucose formation from non-sugar substrates (gluconeogenesis), i.e., phosphoenolpyruvate carboxyl kinase (PEPCK) and glucose-6-phosphatase (G6Pase) [60]. It was also shown that in HFD-fed mice, baicalin therapy limited Sirt1 induction due to a diminishing NAD^+^ pool and thus protected STAT3 activation by preserving acetylation [90]. These changes effectively contribute to reducing hepatic glucose output. This is a relevant finding in the mechanism of the anti-diabetic action of baicalin. Exaggerated hepatic glucose output is one of the significant factors leading to hyperglycemia in diabetic states. Under normal conditions, hepatic glucose production is suppressed by insulin. However, in insulin resistance, this process is not effectively limited, and the liver releases glucose into the blood in spite of hyperglycemia. Thus, the limitation of gluconeogenesis largely contributes to the blood-glucose-lowering effects of baicalin. Baicalin treatment of GK rats was also found to increase citrate synthase activity and mitochondria number in the liver. This indicates a rise in the metabolic capacity of hepatocytes [62].

Insulin resistance is well known to be accompanied by oxidative stress [20,23]. Baicalin supplementation to the insulin-resistant GK rats was shown to upregulate hepatic mRNA and protein expression of antioxidant enzymes, i.e., glutathione peroxidase (GPx), superoxide dismutase (SOD), and catalase (CAT). Importantly, the activity of these enzymes was also increased, while hepatic malondialdehyde (MDA) content diminished. MDA is a marker of lipid peroxidation. Reduced MDA formation along with the augmented activity of GPx, SOD, and CAT indicates the effective antioxidant defense following baicalin therapy [62]. This is a relevant finding since oxidative stress has a role in the pathogenesis and progression of diabetes. Oxidative stress is associated with damage to DNA, lipids, and proteins and, finally, with cell dysfunction [94]. Treatment of patients with type 2 diabetes with ani-oxidants alone is not sufficient to stop the disease but may limit some complications [94].

In animal models of diabetes, baicalin therapy was shown to have hepatoprotective action. Liver damage is associated with increased activities of aspartate aminotransferase (AST) and alanine aminotransferase (ALT). This is followed by the excessive release of these enzymes into the blood and their increased activity. In mice fed an HFD, baicalin reduced the blood activity of AST and ALT [59,68]. These effects were confirmed in insulin-resistant GK rats subjected to baicalin therapy [62]. In the liver tissue, diabetes-related oxidative stress is accompanied by inflammatory stress, which additionally worsens pathological changes. Baicalin therapy was demonstrated to reduce hepatic interleukin content (IL-1, IL-6, and IL-10) in HFD-fed rats [91]. The hepatoprotective action of baicalin contributes to better insulin sensitivity. Under physiological conditions, insulin action at the cellular level involves multiple steps and signaling molecules. The molecular background of insulin resistance in diabetes is also complex, and various stages of its action may be disturbed. The first step of insulin signaling is hormone binding to the membrane receptor. It was shown that in HFD-fed mice, baicalin upregulated the expression of the hepatic insulin receptor. Moreover, baicalin elevated the expression level of pAkt and pAS160. These intracellular proteins are involved in the signal transduction of insulin. These results show that baicalin therapy improves insulin signaling [64]. The mechanism thereby baicalin alleviates diabetes-related liver dysfunction is pleiotropic (Figure 4, Table 4).

### 3.6. Effects of Baicalin on Oxidative and Inflammatory Stress

Reactive oxygen species (ROS) comprise superoxide anion (O_2_^−^), hydrogen peroxide (H_2_O_2_), and hydroxyl radical (OH^−^). The main source of intracellular ROS is the electron transport chain. It is known that mitochondrial ROS formation strongly depends on cellular glucose supply. Under physiological conditions, when blood glucose levels are maintained in the physiological range, tissue ROS formation is relatively low, and there is a balance between their generation and removal. It is estimated that no more than 2% of the total mitochondrial oxygen is converted to superoxide anion and hydrogen peroxide. However, in the case of hyperglycemia, ROS generation rises markedly beyond the normal range and may reach even 10% of the total amount of oxygen passing through the mitochondrial transport chain. Increased ROS production leads to oxidative stress. A relationship between hyperglycemia and augmented ROS formation in diabetic states is well established. Moreover, ROS are thought to be an etiologic factor involved in insulin resistance [20,23,50,94,95,96].

The efficacy of the elimination of ROS is pivotal to preserving cell function and survival. Therefore, cells possess an antioxidant defense system comprising non-enzymic and enzymic components. Enzymes involved in ROS degradation are SOD, CAT, GPx, and glutathione reductase (GR). SOD catalyzes the conversion of O_2_ to H_2_O_2_, CAT is involved in the decomposition of H_2_O_2_ into two products, H_2_O and O2, GPx catalyzes the decomposition of H_2_O_2_, and GR participates in the reduction of GSSG to GSH.

Baicalin was shown to improve blood levels of diabetes-related oxidative stress parameters. Its effectiveness was confirmed in various animal models of diabetes. In rats with diabetes evoked by the administration of STZ, baicalin reduced blood levels of MDA [46], the end product of ROS-induced lipid peroxidation. Moreover, in rats with HFD/HSD and STZ-induced diabetes, baicalin supplementation increased the blood activity of SOD. This effect was accompanied by reduced blood MDA content [57,91]. It was also demonstrated that in the insulin-resistant GK rat, baicalin therapy diminished plasma lipid peroxide and carbonyl contents [62]. These results provided evidence that baicalin is effective in alleviating diabetes-related oxidative stress.

In diabetes, oxidative stress is associated with potentiating inflammatory processes. It is thought that diabetes-related chronic hyperglycemia is the main causative factor for both [20,23,50,95]. In addition, blood-glucose-lowering therapies mitigate oxidative and inflammatory stress. Elevated blood glucose levels are accompanied by inflammatory markers and pro-inflammatory cytokines. These inflammation-related markers are generated in various tissues, contributing to their damage. Moreover, these cytokines are released into the blood [96]. Baicalin therapy was demonstrated to exert anti-inflammatory action since it reduced blood levels of inflammation-related markers. Such effects were shown in rats with diabetes induced by HFD/HSD and STZ, in which baicalin reduced blood levels of inflammatory markers, TNF-α, and interleukin 6 (IL-6) [57]. Moreover, in rats on an HFD, baicalin diminished TNF-α content [56].

Chronically elevated blood glucose levels in diabetes are associated with an exaggerated cellular glucose supply and with the resulting intracellular ROS formation. ROS are very active and induce damage to DNA, cellular proteins, and lipids. This evokes the progressive dysfunction of many cells and tissues [94]. Baicalin may alleviate diabetes-related oxidative stress via three mechanisms. Given the link between elevated blood glucose and oxidative stress, the most probable action of baicalin is to ameliorate oxidative stress in diabetes by decreasing hyperglycemia. The anti-hyperglycemic effects of baicalin have been shown in animals with various models of diabetes. Baicalin may also directly activate antioxidant enzymes, thereby promoting ROS degradation. Another possibility is action as a ROS scavenger by the direct interaction of baicalin with generated ROS. Progressive hyperglycemia leads to persistent inflammation and up-regulates markers of chronic inflammation. Moreover, the inflammatory stress is potentiated by ROS. This indicates the link between diabetic hyperglycemia and oxidative and inflammatory stress [95,96]. Anti-hyperglycemic effects of baicalin seem to be the main causative factor contributing to the alleviation of both oxidative and inflammatory stress.

## 4. Effects of Baicalin on Body Mass

Feeding rodents an HFD is associated with hormonal and metabolic disturbances, partially similar to changes developing in humans with type 2 diabetes. However, these alterations differ depending on experimental conditions, such as the term of ingestion, fat content in a diet, animal strain, and others [97]. HFD may raise body weight gain and induce obesity, which is associated with insulin resistance. Baicalin was demonstrated to reduce body weight gains in mice [58,60,74] and rats [56] fed an HFD. Some studies also revealed that its supplementation in mice on an HFD increases energy expenditure [58]. Leptin is the main hormone regulating food intake, body weight, and energy expenditure. This hormone is an adipocyte-derived satiety signal, reaches the hypothalamus, and limits food intake. However, in obesity, blood leptin levels are permanently elevated, which is associated with leptin resistance. Leptin resistance concerns not only the hypothalamus but also peripheral, metabolically active tissues. On the other hand, reduced fat tissue content and the resulting normalization of blood leptin levels recover the physiological action of this hormone [87]. Thus, it is possible that the baicalin action in HFD-induced obese mice and rats is partially mediated by leptin.

The majority of studies have shown that baicalin therapy reduced HFD-induced obesity and in that way improved insulin sensitivity and ameliorated other diabetes-related disturbances [56,58,59,60,74]. However, other results provided evidence that this compound can alleviate diabetes-related parameters without detectable changes in body weight [68]. Moreover, baicalin was also effective in GK rats, a non-obese model of type 2 diabetes [62]. Consumption of an HFD may induce insulin resistance and other diabetes-related disturbances without a detectable rise in body weight. In this case, pathological changes are due to increased liver lipid accumulation, elevated ectopic fat storage in the skeletal muscle, and adipocyte dysfunction. Moreover, it has been well proven that HFD evokes epigenetic modifications, such as DNA methylation, histone modifications, and the expression of non-coding RNAs [97]. These changes are associated with hormonal and metabolic disturbances; however, body weight may be unaltered. This indicates that baicalin alleviates diabetes-related disturbances by inducing molecular changes at the cellular level, and a decrease in body weight is not a prerequisite for its action.

## 5. Effective Doses and Toxicity

In rodents with diet-induced obesity and insulin resistance, baicalin was usually given at a dose of 100 or 200 mg/kg b.w. [57,59]. The lowest effective dose was 25 mg [61]. The maximal effective amount of baicalin given to rats and mice was 400 mg per kg b.w. [58,59]. The liver is the main organ responsible for detoxification reactions. In the case of liver injury, some enzymes, especially aminotransferases, are released into the blood, increasing their blood activities. It was shown that baicalin was given for 14 weeks to mice at a dose of 400 mg per kg b.w. and did not induce changes in blood ALT and AST activities [59]. Moreover, no side effects were observed in rats receiving baicalin at a dose of 400 mg per kg b.w for 12 weeks [58]. This indicates that long-term baicalin ingestion is safe and non-toxic. Moreover, this compound can protect organs and tissues against various toxins [6,98]. This is a relevant feature since some compounds may have contradictory action, alleviating diabetes-related changes at lower doses and aggravating them at higher doses [99,100].

Results of studies on various animal models of diabetes provided evidence that the efficacy of baicalin is comparable to metformin action. Metformin is a biguanide drug commonly used in humans with type 2 diabetes [20,101]. It was shown that baicalin given to insulin-resistant GK rats at a dose of 120 mg/kg b.w. was more effective in alleviating many diabetes-related disturbances than 500 mg metformin [62]. Moreover, in NA/STZ rats, treatment with 100 mg baicalin or 100 mg metformin evoked comparable changes in mitigating diabetes [52]. Studies on HFD-fed mice revealed that baicalin given at doses of 70 or 210 mg/kg b.w. and 200 mg metformin caused similar beneficial alterations [102].

## 6. Conclusions and Additional Remarks

Baicalin was shown to alleviate diabetes-related disturbances in rats and mice with experimentally induced diabetes. Diabetes is a complex disease, and the function of many tissues is seriously impaired. Hyperglycemia and impaired insulin action are the main hallmarks of diabetes. It was demonstrated that baicalin was effective in suppressing hyperglycemia and improving insulin action. Baicalin therapy was associated with beneficial changes covering the main insulin-sensitive tissues: the skeletal muscle, the adipose tissue, and the liver.

In the muscle tissue, baicalin was shown to reduce lipid accumulation. This effect resulted from alterations in phosphorylation and the action of enzymes involved in the regulation of lipid metabolism. Baicalin therapy was associated with phosphorylation of AMPKα at the activation site of the enzyme. Activated AMPK induced phosphorylation of ACC at the inhibitory site. Moreover, baicalin treatment was demonstrated to reduce the skeletal muscle content of a key signaling molecule involved in the regulation of lipid metabolism, i.e., malonyl-CoA. All these changes promoted the mitochondrial oxidation of FAs and thereby diminished lipid accumulation. Baicalin increased the phosphorylation of intracellular molecules related to insulin signaling, PKB/Akt, and GSK-3β and also up-regulated the expression of PGC-1α, p38MAPK, and p-AS160. It was also demonstrated that baicalin elevated the expression level of the insulin-regulated glucose transporter GLUT4. These changes indicate that baicalin improved insulin sensitivity in the muscle tissue and increased intramuscular glucose transport. Given that the skeletal muscle is responsible for the main glucose uptake, this effect contributes to the blood-glucose-lowering action of baicalin. Baicalin was also demonstrated to diminish adipose tissue content. This effect was due to increased AMPK-dependent mitochondrial biogenesis and the resulting rise in metabolic activity. Reduced adipose tissue content was associated with decreased inflammatory markers and macrophage accumulation. These effects contribute to better adipose tissue insulin action. The beneficial influence of baicalin on insulin action shown in the liver tissue was largely due to reduced lipid accumulation. This resulted from both increased FA degradation and reduced FA synthesis. The former was due to AMPK activation and ACC inhibition and also reduced malonyl-CoA content. This eliminated the inhibitory effect of malonyl-CoA on FAs transport to mitochondria and their oxidation; moreover, it was confirmed that baicalin increased both CPT1A activity and β-oxidation. Baicalin therapy also increased the expression of hepatic glucose transporters (GLUT1 and GLUT4) and reduced the expression of enzymes involved in glucose formation (PEPCK and G6Pase). This indicates that baicalin increased glucose uptake and simultaneously reduced glucose release. These changes largely contribute to the blood-glucose-lowering action.

The effectiveness of baicalin action has been revealed in various animal models of diabetes. This suggests the large therapeutic potential of baicalin to support conventional therapies in patients with diabetes. However, human studies are required to assess the real therapeutic value of baicalin in alleviating diabetes-related disturbances. Given that diabetes is a chronic disease, long-term treatment with anti-diabetic drugs is required [103,104]. The hormonal and metabolic disturbances in people with diabetes are associated with grave diabetes complications [105,106]. Therefore, research concerning the anti-diabetic effects of baicalin should take into account all these aspects. Based on the results of animal studies, the effective doses of baicalin in humans should be determined. Moreover, human studies are required to determine the potential side effects of long-term baicalin therapy.

Another important issue is to determine whether baicalin exerts any interactions with drugs, especially those used in diabetes. It is known that many natural-derived compounds markedly influence the pharmacokinetics of drugs and may also reinforce or attenuate their effectiveness [107,108,109].

Baicalin absorption from the digestive tract is relatively low because of its metabolism by the intestinal microbiota. Therefore, the bioavailability of the parent compound to organs and cells is relatively slight [110]. The form of baicalin with a stabilized structure, less susceptible to intestinal transformations, should be prepared and tested in humans. Minor degradation could result in higher anti-diabetic activity at lower doses [111,112].

It is well documented that in animal studies, the effectiveness of many natural, biologically active compounds changes. Different doses may induce various effects, and the mechanism of action at different doses may also vary. This is mainly due to differences in experimental conditions [40,112,113,114]. In the case of human research, the results are usually less coherent and conclusive. This results from differences in experimental conditions, a small sample size, and the short-term duration of the research [40,115]. Therefore, human studies on the anti-diabetic effects of baicalin should be performed over a long-term period and involve a significant number of volunteers. Obtaining positive results in preliminary studies would enable the use of baicalin as an anti-diabetic drug to effectively support conventional therapies in humans.

## Figures and Tables

**Figure 1 ijms-25-00431-f001:**
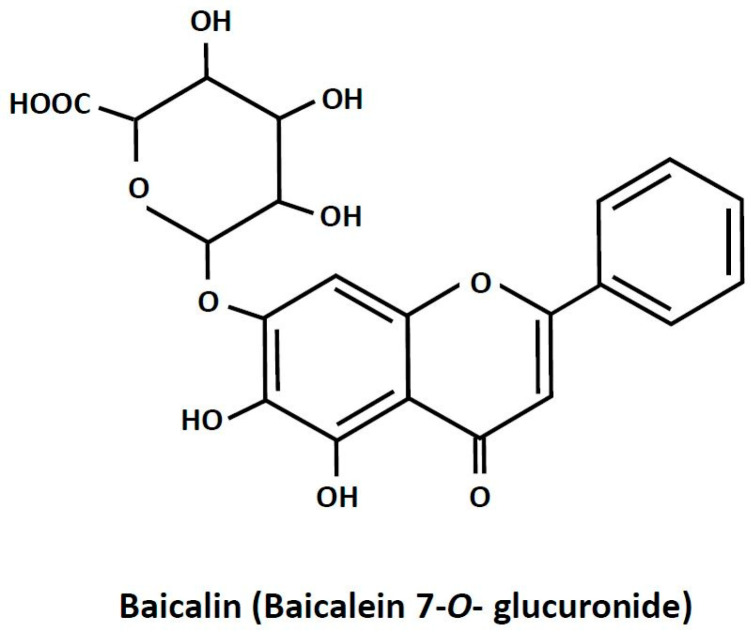
The chemical structure of baicalin.

**Figure 2 ijms-25-00431-f002:**
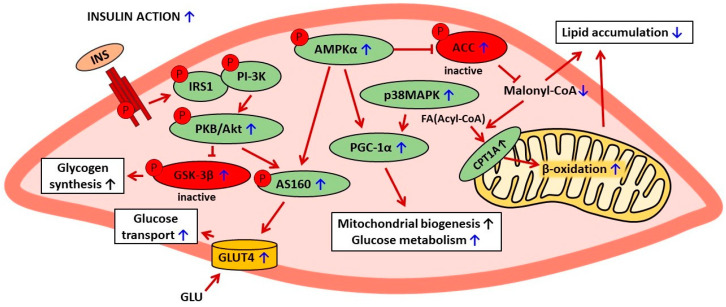
The suggested mechanisms of baicalin action in skeletal muscle cells in rodent models of metabolic disorders. The confirmed effects of baicalin are marked with a blue arrow, the suggested effects are marked with a black arrow. AMPKα—AMP-activated protein kinase (α subunit), ACC—acetyl-CoA carboxylase, CPT1A—carnitine palmitoyltransferase 1A, FA—fatty acid, INS—insulin, IRS1—insulin receptor substrate 1, PI-3K—phosphatidylinositol 3 kinase, PKB/Akt—protein kinase B, GSK-3β—glycogen synthase kinase-3β, PGC-1α—peroxisome proliferator-activated receptor-gamma coactivator 1 α, p38MAPK—p38 mitogen-activated protein kinase, p-AS160—phosphorylated Akt substrate of 160 kDa, GLU—glucose, GLUT4—glucose transporter 4.

**Figure 3 ijms-25-00431-f003:**
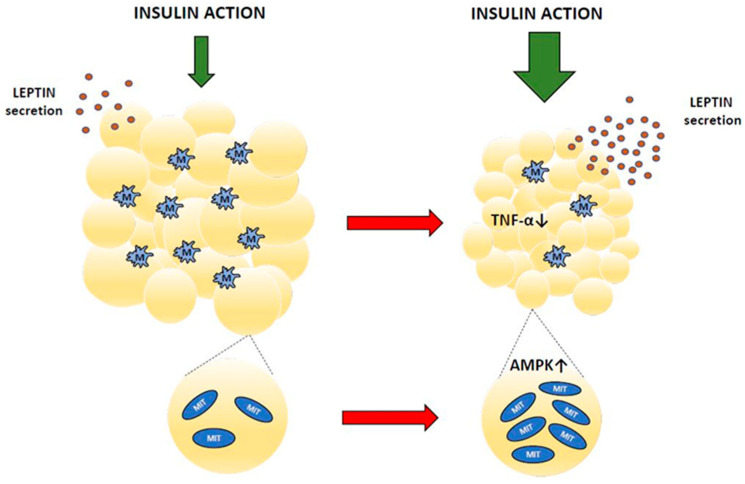
Adipose tissue changes after baicalin action (red arrow means baicalin action). MIT—mitochondrium, M—macrophage, TNF-α—tumor necrosis factor α, AMPK—AMP-activated protein kinase.

**Figure 4 ijms-25-00431-f004:**
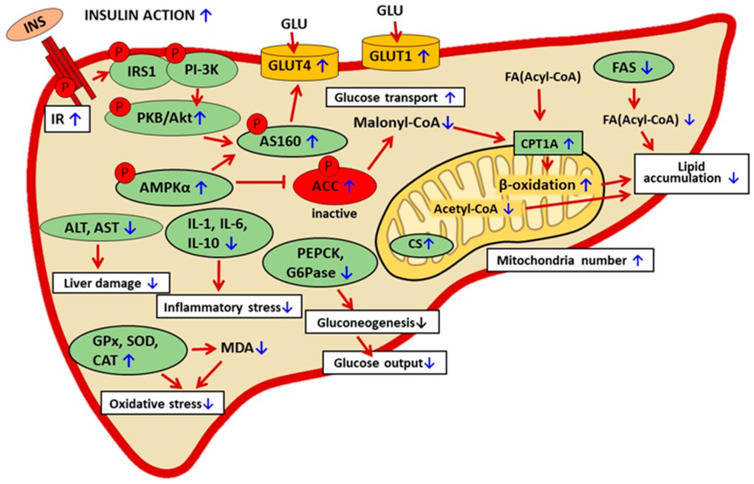
The suggested mechanisms of baicalin action in liver cells in rodent models of metabolic disorders. The confirmed effects of baicalin are marked with a blue arrow, the suggested effect is marked with a black arrow. INS—insulin, IR—insulin receptor, INS—insulin, IRS1—insulin receptor substrate 1, PI-3K— phosphatidylinositol 3 kinase, AMPKα—AMP-activated protein kinase (α subunit), ACC—acetyl-CoA carboxylase, FA—fatty acid, PKB/Akt—protein kinase B, p-AS160—phosphorylated Akt substrate of 160 kDa, CPT1A—carnitine palmitoyltransferase 1A, CS—citrate synthase, GLU—glucose, GLUT 1—glucose transporter 1, GLUT4—glucose transporter 4, PEPCK—phosphoenolpyruvate carboxykinase, G6Pase—glucose 6-phosphatase, GPx—glutathione peroxidase, SOD—superoxide dismutase, CAT—catalase, MDA—malondialdehyde, AST—aspartate aminotransferase, ALT—alanine aminotransferase, IL-1, IL-6, IL-10—interleukin 1, 6, 10.

**Table 1 ijms-25-00431-t001:** The effects of baicalin on blood glucose levels and insulin resistance.

Parameter	Effect	Animal Model	References
**Hyperglycemia**	Decrease	rats with STZ-induced diabetes;	[42,43,44,46]
		mice with STZ-induced diabetes;	[45]
		rats with STZ-NA-induced diabetes;	[52]
		rats with STZ-induced diabetes;	[51]
		HFD rats;	[56,57]
		HFD/HSD/STZ rats;	[58,59]
		HFD mice;	[60,61]
		GK rats;	[62]
**Insulin resistance**	Alleviation	HFD mice;	[51,60,61,64]
		GK rats;	[62]
		db/db mice;	[65]
**Hyperinsulinemia**	Decrease	HFD mice;	[59,60,64]
		HFD rats;	[56]
		GK rats;	[62]
**Hypoleptinemia**	Increase	GK rats;	[62]
		rats with STZ-induced diabetes;	[42,51]

STZ—streptozotocin, NA—nicotinamide, HFD—high-fat diet, HSD—high-sucrose diet, GK rats—Goto-Kakizaki rats.

**Table 2 ijms-25-00431-t002:** The effects of baicalin on the skeletal muscle.

Parameter	Effect	Possible Way of Action		Animal Model	References
Insulin resistance	Alleviation	Blood NEFAs	↓	Insulin-resistant mice; HFD rats; HFD/HSD/STZ rats;	[57,59,64,65]
		Blood TGs	↓	HFD, rats; GK rats; obese mice;	[56,59,62,68]
Lipid accumulation	Decrease	AMPKα (at Thr172) phosphorylation;	↑	HFD, mice	[59]
		ACC phosphorylation (at Ser79) and ACC inhibition;	↑		
		malonyl-CoA synthesis;	↓		
		FAs oxidation	↑		
Glucose transport and metabolism	Improvement	PKB/Akt phosphorylation (at Thr308);	↑	HFD, mice	[59]
		GSK-3β phosphorylation (at Ser9);	↑		
		PGC-1α, p38MAPK, p-AS160 expression	↑		
		GLUT4 expression	↑	obese mice	[61,74]
		Galanin resistance (blood galanin) ↓	↓	obese mice	[61]

NEFAs—non-esterified fatty acids, HFD—high-fat diet, HSD—high sucrose diet, STZ—streptozotocin, GK rats—Goto-Kakizaki rats, AMPKα—AMP-activated protein kinase (α subunit), ACC—acetyl-CoA carboxylase, TGs—triglycerides, FAs—fatty acids, PKB/Akt—protein kinase B, GSK-3β—glycogen synthase kinase-3β, PGC-1α—peroxisome proliferator-activated receptor-gamma coactivator 1 α, p38MAPK—p38 mitogen-activated protein kinase, p-AS160—phosphorylated Akt substrate of 160 kDa, GLUT4—glucose transporter 4. Red and black arrow means decrease; green arrow means increase.

**Table 3 ijms-25-00431-t003:** The effects of baicalin on adipose tissue.

Parameter	Effect	Possible Way of Action		Animal Model	References
Body fat content	Reduced	Insulin sensitivity	↑	HFD rats HFD mice	[56,68,78]
Inflammatory stress	Alleviation	Macrophage content	↓	HFD, mice	[64]
		TNF-α expression	↓		
Insulin resistance	Alleviation	AMPK-dependent, mitochondrial biogenesis	↑	HFD, mice	[83]

HFD—high-fat diet, STZ—streptozotocin, GK rats—Goto-Kakizaki rats, AMPK—AMP-activated protein kinase, TNF-α—tumor necrosis factor α. Red arrow means decrease; green arrow means increase.

**Table 4 ijms-25-00431-t004:** The effects of baicalin on the liver.

Parameter	Effect	Possible Way of Action		Animal Model	References
Insulin resistance	Alleviation	Lipid accumulation;	↓	HFD mice	[58,59]
				HFD rats	[56,91]
		Expression of IR, pAkt, pAS160;	↑	HFD mice	[64]
Lipid accumulation	Decrease	AMPKα (at Thr172) phosphorylation;	↑	HFD mice	[59]
		ACC phosphorylation (at Ser79) and inhibition;	↑	HFD rats	[56]
		malonyl-CoA synthesis;	↓		
		FAs oxidation;	↑		
		Acetyl-CoA;	↓	HFD mice	[90]
		FAs oxidation;	↑		
		CPT1A activity;	↑	HFD mice	[58]
		FAs oxidation;	↑		
		FAS expression;	↓	HFD rats	[56]
		FAs synthesis;	↓		
Glucose transport	Improvement	GLUT1 and GLUT4 expression;	↑	HFD mice	[60]
Glucose output	Decrease	PEPCK and G6Pase expression;	↓	HFD mice	[60]
		NAD^+^ pool;	↓	HFD mice	[90]
		Sirt1 induction;	↓		
		Preservation STAT3 acetylation	↑		
Metabolic capacity	Increase	Citrate synthase activity;	↑	GK rats	[62]
		Mitochondria number	↑		
Oxidative stress	Decrease	Antioxidant enzymes’ protein expression (GPx, SOD, and CAT);	↑	GK rats	[62]
		MDA content	↓		
Liver damage	Decrease	Activity of AST and ALT	↓	HFD mice,	[59,68]
				GK rats	[62]
Inflammatory stress	Decrease	Interleukin IL-1, IL-6, and IL-10 content	↓	HFD rats	[91]

HFD—high-fat diet, GK rats—Goto-Kakizaki rats, IR—insulin receptor, AMPKα—AMP-activated protein kinase (α subunit), ACC—acetyl-CoA carboxylase, FAs—fatty acids, pAkt— phosphorylated protein kinase B, p-AS160—phosphorylated Akt substrate of 160 kDa, CPT1A—carnitine palmitoyltransferase 1A, GLUT 1—glucose transporter 1, GLUT4—glucose transporter 4, PEPCK—phosphoenolpyruvate carboxykinase, G6Pase—glucose 6-phosphatase, Sirt1—NAD-dependent protein deacetylase, STAT3—signal transducer and activator of transcription 3, GPx—glutathione peroxidase, SOD—superoxide dismutase, CAT—catalase, MDA—malondialdehyde, AST—aspartate aminotransferase, ALT—alanine aminotransferase, IL-1, IL-6, IL-10—interleukin 1, 6, 10. Red arrow means decrease; green arrow means increase.

## Data Availability

Not applicable.

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
