# Peer review of "The Anti-Diabetic Potential of Baicalin: Evidence from Rodent Studies"

_ijms, 2023, doi:10.3390/ijms25010431_

Round 1
Reviewer 1 Report
Comments and Suggestions for Authors
This submission is a review paper, the research progress on the anti-diabetes effect of baicalin in rodents was summarized by this review paper. However, the manuscript should be improved.
1. The abstract is too concise and relevant key information should be supplemented.
2. Provide five suitable keywords.
3. The theme of the article is anti-diabetes, but when the author discussed separately, they explained the role of baicalin in different tissues, oxidative stress, and inflammatory injury, and the oxidative stress and inflammatory injury are also repeated in different tissues, what is the relationship between these parts and anti-diabetes? In other words, the structure of the article is not consistent with the theme, and the author must rewrite and reorganize the manuscript.
4. The title claimed that the studies from rodents, but here only rats and mice, no studies related to rabbits?
5. As a comprehensive review, the lack of figures throughout the text is really inappropriate, supplement some figures that summarize existing researches or mode of action, especially the molecular or biochemical mechanisms of the action of baicalin. Otherwise, this article is not suitable for publication in high IF journals like IJMS.
6. The style of the tables should refer to the requirements of the journal.
7. For a high-quality review, the author needs to consult more literature.
8. The conclusion should be more specific, describing the specific efficacy and mechanism of baicalin in anti diabetes, its application prospect, existing problems, how to solve these problems in the future and what kind of research should be carried out to solve these problems. On the basis of summarizing these literatures, the author should add their own in-depth views.
9. Moderate editing of English language is required.
10. In summary, although this review is comprehensive, there are many issues, and the author must significantly improve the quality of the manuscript to meet the requirements for publication on IJMS.
Comments on the Quality of English LanguageModerate editing of English language required
Author Response
Reviewer 1
Thank you very much for your valuable comments and suggestions. Our manuscript has been markedly improved, and all your suggestions have been taken into account.
This submission is a review paper, the research progress on the anti-diabetes effect of baicalin in rodents was summarized by this review paper. However, the manuscript should be improved.
- The abstract is too concise and relevant key information should be supplemented.
According to your suggestions, the abstract has been corrected and important information related to the anti-diabetic effects of baicalin added:
Baicalin is a biologically active flavonoid compound that benefits the organism in various pathological conditions. Rodent studies have shown that this compound effectively alleviated diabetes-related disturbances in models of type 1 and type 2 diabetes. Baicalin supplementation limited hyperglycemia and improved insulin sensitivity. The anti-diabetic effects of baicalin covered the main insulin-sensitive tissues, i.e., the skeletal muscle, the adipose tissue, and the liver. In the muscle tissue, baicalin limited lipid accumulation and improved glucose transport. Baicalin therapy was associated with diminished adipose tissue content and increased mitochondrial biogenesis. Hepatic lipid accumulation and glucose output were also decreased as a result of baicalin supplementation. The molecular mechanism of anti-diabetic action of this compound is pleiotropic and is associated with the changes in the expression/action of pivotal enzymes and signaling molecules. Baicalin positively affected among others tissue insulin receptor, glucose transporter, AMP-activated protein kinase, protein kinase B, carnitine palmitoyltransferase, acetyl-CoA carboxylase, and fatty acid synthase. Moreover, this compound ameliorated diabetes-related oxidative and inflammatory stress and reduced epigenetic modifications. Importantly, baicalin supplementation at the effective doses did not induce any side effects. Results of rodent studies imply that baicalin may be tested as an anti-diabetic agent in humans.
- Provide five suitable keywords.
According to Reviewer suggestions, five suitable keywords have been added:
hyperglycemia, metabolic dysregulation, skeletal muscle, adipose tissue, liver
- The theme of the article is anti-diabetes, but when the author discussed separately, they explained the role of baicalin in different tissues, oxidative stress, and inflammatory injury, and the oxidative stress and inflammatory injury are also repeated in different tissues, what is the relationship between these parts and anti-diabetes? In other words, the structure of the article is not consistent with the theme, and the author must rewrite and reorganize the manuscript.
Reviewer suggestion concerning the reorganization of our manuscript is fully adequate. However, the antioxidant defense system is different in each tissue. This is mainly due to differences in the expression/activity of antioxidant enzymes. Moreover, the susceptibility of various tissues to inflammatory stress is also different as a result of different expressions of inflammation-related factors. Therefore, we suggest mentioning in the text the oxidative and inflammatory stress, describing each tissue: pancreatic β-cells, the skeletal muscle, the adipose tissue, and the liver. (Otherwise, some data would have to be repeated when describing each tissue). However, general information concerning the diabetes-related oxidative and inflammatory stress, and also concerning the action of baicalin is included and expended in the part of our manuscript baicalin: “effects of baicalin on oxidative and inflammatory stress.”
Oxidative and inflammatory stress is strongly associated with the pathogenesis and progression of diabetes and its complications. Some data on the link between diabetes, oxidative, and inflammatory stress are provided in chapters concerning pancreatic β-cells, the skeletal muscle, the adipose tissue, and the liver. The examples are listed below:
“Sustained hyperglycemia leads to increased generation of reactive oxygen species (ROS), which additionally impairs β-cell structure and endocrine functions [23]. Compared with other kinds of cells, pancreatic β-cells are known to be very susceptible to the action of ROS. This is due to a weak anti-oxidant defense system [49,50].
Insulin resistance is well-known to be accompanied by oxidative stress [20,23]. Baicalin supplementation to the insulin-resistant GK rats was shown to upregulate hepatic mRNA and protein expression of antioxidant enzymes, i.e. glutathione peroxidase (GPx), superoxide dismutase (SOD), and catalase (CAT). Importantly, the activity of these enzymes was also increased, while hepatic malondialdehyde (MDA) content diminished. MDA is a marker of lipid peroxidation. Reduced MDA formation along with the augmented activity of GPx, SOD, and CAT indicates the effective anti-oxidant defense following baicalin therapy [62]. This is a relevant finding since oxidative stress has a role in the pathogenesis and progression of diabetes. Oxidative stress is associated with damage to DNA, lipids, and proteins and, finally with cell dysfunction [94]. Treatment of patients with type 2 diabetes with ani-oxidants alone is not sufficient to stop the disease but may limit some complications [94].
Obesity is associated with pathological adipose tissue remodeling. Among various obesity-related changes, immune cell infiltration has relevant implications. Macrophages, which accumulate within adipose tissue, secrete inflammatory cytokines. This evokes chronic low-grade inflammation and inflammatory stress. Inflammatory stress is one of the relevant factors contributing to impaired insulin action, insulin resistance, and type 2 diabetes [20,79,80,81]. Baicalin therapy was shown to alleviate these unfavorable changes effectively. It was demonstrated that in mice on an HFD, baicalin decreased adipose tissue content, reduced tissue macrophage content, and down-regulated expression of the pro-inflammatory cytokine, tumor-necrosis factor-α (TNF-α) [64]. This indicates that baicalin has an anti-inflammatory action in the adipose tissue of obese mice. The anti-inflammatory effects of baicalin were shown in many other pathological conditions [1,9].
In the liver tissue, diabetes-related oxidative stress is accompanied by inflammatory stress, which additionally worsens pathological changes. Baicalin therapy was demonstrated to reduce hepatic interleukin content (IL-1, IL-6, and IL-10) in HFD-fed rats [91].
A relationship between hyperglycemia and augmented ROS formation in diabetic states is well established. Moreover, ROS are thought to be an etiologic factor involved in insulin resistance.”
According to Reviewer suggestions, more information linking oxidative and inflammatory stress and diabetes has been expanded (in the chapter: Effects of baicalin on oxidative and inflammatory stress):
Chronically elevated blood glucose levels in diabetes are associated with an exaggerated cellular glucose supply and with the resulting intracellular ROS formation. ROS are very active and induce damage to DNA, cellular proteins, and lipids. This evokes progressive dysfunction of many cells and tissues [94]. Baicalin may alleviate diabetes-related oxidative stress via three mechanisms. Given the link between elevated blood glucose and oxidative stress, the most probable action of baicalin is to ameliorate oxidative stress in diabetes by decreasing hyperglycemia. The antihyperglycemic effects of baicalin have been shown in animals with various models of diabetes. Baicalin may also directly activate antioxidant enzymes thereby promoting ROS degradation. Another possibility is action as a ROS scavenger, via direct interaction of baicalin with generated ROS. Progressive hyperglycemia leads to persistent inflammation and up-regulates markers of chronic inflammation. Moreover, the inflammatory stress is potentiated by ROS. This indicates the link between diabetic hyperglycemia, and oxidative and inflammatory stress [95,96]. Anti-hyperglycemic effects of baicalin seem to be the main causative factor contributing to the alleviation of both oxidative and inflammatory stress.
- The title claimed that the studies from rodents, but here only rats and mice, no studies related to rabbits?
It would be interesting to expand our manuscript on data concerning the anti-diabetic effects of baicalin in rabbits. However, rabbits are much less frequently used in studies on diabetes than rats and mice. In rodents, diabetes is easily induced by the administration of streptozotocin, by feeding a high-energy diet, or by genetic manipulation. These rodents are very suitable models for type 1 and type 2 diabetes, especially in the context of studying diabetes-related hormonal and metabolic disturbances. Therefore, rats and mice are commonly used in research on anti-diabetic action of new drugs, and natural compounds. Moreover, literature data on the anti-diabetic effects of baicalin in rabbits is lacking.
- As a comprehensive review, the lack of figures throughout the text is really inappropriate, supplement some figures that summarize existing researches or mode of action, especially the molecular or biochemical mechanisms of the action of baicalin. Otherwise, this article is not suitable for publication in high IF journals like IJMS.
According to Reviewer suggestions, Figures summarizing the known, conformed by literature data, anti-diabetic effects of baicalin have been added. The biochemical mechanisms of anti-diabetic baicalin action are shown. This indeed markedly raises the value of our manuscript.
We tried to use all available data provided by the authors. However, some mechanisms of baicalin action are not fully explored and further research is needed. For example, Dai et al., (2018) have shown that baicalin is a competitive activator of CPT1A and thereby promotes FAs oxidation. However, these authors did not study the exact mechanism by which baicalin affects this enzyme.
- The style of the tables should refer to the requirements of the journal.
In accordance with the IJMS instructions, a template from Microsoft Word was used to prepare the tables - table design. Arrows that were previously included as shapes have been removed from the tables - they have been replaced with arrows from the font.
- For a high-quality review, the author needs to consult more literature.
According to Reviewer suggestions, more literature data have been included. Now, all available literature data related to the topic of our manuscript have been taken into account.
(see below)
- The conclusion should be more specific, describing the specific efficacy and mechanism of baicalin in anti diabetes, its application prospect, existing problems, how to solve these problems in the future and what kind of research should be carried out to solve these problems. On the basis of summarizing these literatures, the author should add their own in-depth views.
According to Reviewer suggestions, the conclusion has been markedly expanded and new data added:
Conclusion and additional remarks
Baicalin was shown to alleviate diabetes-related disturbances in rats and mice with experimentally induced diabetes. Diabetes is a complex disease, and the function of many tissues is seriously impaired. Hyperglycemia and impaired insulin action are the main hallmarks of diabetes. It was demonstrated that baicalin was effective in suppressing hyperglycemia and improving insulin action. Baicalin therapy was associated with beneficial changes covering the main insulin-sensitive tissues, the skeletal muscle, the adipose tissue, and the liver.
In the muscle tissue, baicalin was shown to reduce lipid accumulation. This effect resulted from alterations in phosphorylation and the action of enzymes involved in the regulation of lipid metabolism. Baicalin therapy was associated with phosphorylation of AMPKα at the activation site of the enzyme. Activated AMPK induced phosphorylation of ACC at the inhibitory site. Moreover, baicalin treatment was demonstrated to reduce the skeletal muscle content of a key signaling molecule involved in the regulation of lipid metabolism, i.e. malonyl-CoA. All these changes promoted the mitochondrial oxidation of FAs and thereby diminished lipid accumulation. Baicalin increased the phosphorylation of intracellular molecules related to insulin signaling, PKB/AKT, and GSK-3β and also up-regulated the expression of PGC-1α, p38MAPK, and p-AS160. It was also demonstrated that baicalin elevated the expression level of insulin-regulated glucose transporter GLUT4. These changes indicate that baicalin improved insulin sensitivity in the muscle tissue and increased intramuscular glucose transport. Given that the skeletal muscle is responsible for the main glucose uptake, this effect contributes to the blood-glucose-lowering action of baicalin. Baicalin was also demonstrated to diminish adipose tissue content. This effect was due to increased AMPK-dependent mitochondrial biogenesis and the resulting rise in metabolic activity. Reduced adipose tissue content was associated with decreased inflammatory markers and macrophage accumulation. These effects contribute to better adipose tissue insulin action. The beneficial influence of baicalin on insulin action shown in the liver tissue was largely due to reduced lipid accumulation. This resulted from both increased FAs degradation and reduced FAs synthesis. The former was due to AMPK activation and ACC inhibition and also reduced malonyl-CoA content. This eliminated inhibitory effect of malonyl-CoA on FAs transport to mitochondria and their oxidation – moreover it was confirmed that baicalin increased both – CPT1A activity and β-oxidation. Baicalin therapy also increased the expression of hepatic glucose transporters (GLUT1 and GLUT4) and reduced the expression of enzymes involved in glucose formation (PEPCK and G6Pase). This indicates that baicalin increased glucose uptake and simultaneously reduced glucose release. These changes largely contribute to the blood-glucose-lowering action.
The effectiveness of baicalin action has been revealed in various animal models of diabetes. This suggests the large therapeutic potential of baicalin to support conventional therapies in patients with diabetes. However, human studies are required to assess the real therapeutic value of baicalin in alleviating diabetes-related disturbances. Given that diabetes is a chronic disease long-term treatment with anti-diabetic drugs is required [103,104]. The hormonal and metabolic disturbances in people with diabetes are associated with grave diabetes complications [105,106]. Therefore, research concerning the anti-diabetic effects of baicalin should take into account all these aspects. Based on the results of animal studies, the effective doses of baicalin in humans should be determined. Moreover, human studies are required to determine the potential side effects of long-term baicalin therapy.
Another important issue is to determine whether baicalin exerts any interactions with drugs, especially with those used in diabetes. It is known that many natural-derived compounds markedly influence the pharmacokinetics of drugs, and also may reinforce or attenuate their effectiveness [107,108,109].
Baicalin absorption from the digestive tract is relatively low because of its metabolism by intestinal microbiota. Therefore, the bioavailability of the parent compound to organs and cells is relatively slight [110]. The form of baicalin with a stabilized structure, less susceptible to intestinal transformations, should be prepared and tested in humans. Minor degradation could result in higher anti-diabetic activity at lower doses [111,112].
It is well-documented that in animal studies, the effectiveness of many natural, biologically active, compounds changes. Different doses may induce different effects and the mechanism of action at different doses may also vary. This is mainly due to differences in experimental conditions [40,112,113,114]. In the case of human research, the results are usually less coherent and conclusive. This results from differences in experimental conditions, low sample size, and short-term duration of the research [40, 115]. Therefore, human studies on the anti-diabetic effects of baicalin should be performed in the long term and involve a significant number of volunteers. Obtaining positive results in preliminary studies would enable the use of baicalin as an anti-diabetic drug to effectively support conventional therapies in humans.
- Hompesch, M.; Patel, D.K.; LaSalle, J.R.; Bolli, G.B. Pharmacokinetic and pharmacodynamic differences of new generation, longer-acting basal insulins: potential implications for clinical practice in type 2 diabetes. Postgrad. Med. 2019, 131, 117-128.
- Su, J.; Luo, Y.; Hu, S.; Tang, L.; Ouyang. S. Advances in research on type 2 diabetes mellitus targets and therapeutic agents. Int. J. Mol. Sci. 2023, 29, 13381.
- Antar, S.A.; Ashour, N.A.; Sharaky, M.; Khattab, M.; Ashour, N.A.; Zaid, R.T.; Roh, E.J.; Elkamhawy, A.; Al-Karmalawy, A.A. Diabetes mellitus: Classification, mediators, and complications; A gate to identify potential targets for the development of new effective treatments. Biomed. Pharmacother. 2023, 168, 115734.
- Mohamadi, N.; Baradaran Rahimi, V.; Fadaei, M.; Sharifi, F.; Askari, V.R. A mechanistic overview of sulforaphane and its derivatives application in diabetes and its complications. Inflammopharmacol. 2023, 31, 2885-2899.
- Thikekar, A.K.; Thomas, A.B.; Chitlange, S.S. Herb-drug interactions in diabetes mellitus: A review based on pre-clinical and clinical data. Phytother. Res. 2021, 35, 4763-4781.
- Blahova, J.; Martiniakova, M.; Babikova, M.; Kovacova, V.; Mondockova, V.; Omelka, R. Pharmaceutical drugs and natural therapeutic products for the treatment of type 2 diabetes mellitus. Pharmaceuticals (Basel). 2021, 17, 806.
- Zhou, X.; Fu, L.; Wang, P.; Yang, L.; Zhu, X.; Li, CG. Drug-herb interactions between Scutellaria baicalensis and pharmaceutical drugs: Insights from experimental studies, mechanistic actions to clinical applications. Biomed. Pharmacother. 2021, 138, 111445.
- Noh, K.; Kang, Y.; Nepal, M.R.; Jeong, K.S.; Oh, D.G.; Kang, M.J.; Lee, S.; Kang, W.; Jeong, H.G.; Jeong, T.C. Role of intestinal microbiota in baicalin-induced drug interaction and its pharmacokinetics. Molecules 2016, 21,337.
111 .Taghipour, Y.D.; Hajialyani, M.; Naseri, R.; Hesari, M.; Mohammadi, P.; Stefanucci, A.; Mollica, A.; Farzaei, M.H.; Abdollahi, M. Nanoformulations of natural products for management of metabolic syndrome. Int. J. Nanomed. 2019, 16, 5303-5321.
- Dewanjee, S.; Chakraborty, P.; Mukherjee, B.; De Feo, V. Plant-based antidiabetic nanoformulations: the emerging paradigm for effective therapy. Int. J. Mol. Sci. 2020, 21, 2217.
- Dinda, B.; Dinda, S.; DasSharma, S.; Banik, R.; Chakraborty, A.; Dinda, M. Therapeutic potentials of baicalin and its aglycone, baicalein against inflammatory disorders. Eur. J. Med. Chem. 2017, 131, 68-80.
- Rong, J.; Fu, F.; Han, C.; Wu, Y.; Xia, Q.; Du, D. Tectorigenin: A review of its sources, pharmacology, toxicity, and pharmacokinetics. Molecules 2023, 28, 5904.
- Furman, B.L.; Candasamy, M.; Bhattamisra, S.K.; Veettil, S.K. Reduction of blood glucose by plant extracts and their use in the treatment of diabetes mellitus; discrepancies in effectiveness between animal and human studies. J. Ethnopharmacol. 2020, 247, 112264.
- Moderate editing of the English language is required.
English has been verified by a native spiker. Moreover, the whole text has been corrected using the computer program Grammarly.
- In summary, although this review is comprehensive, there are many issues, and the author must significantly improve the quality of the manuscript to meet the requirements for publication on IJMS.
Our manuscript has been markedly changed and improved. We hope that in the present form, the manuscript is more suitable for publication in IJMS.
Reviewer 2 Report
Comments and Suggestions for Authors
The authors successfully highlight the hydrophobic implications of baicalin on membrane lipid metabolism and carbohydrate metabolism. The authors do not clearly extrapolate the correlations between glucose transport at the membrane level and lipid metabolism modulating elements. The authors should try to suggest at least one potential mechanism involved in the successfully identified effects. For example, the existence of a baicalin structure that suggests a rigidity and a modulation of membrane fluidity, may suggest a modulation of the function of lipid shelves. This could justify the large number of metabolic modulations successfully identified by the authors.
Author Response
Thank You very much for Your suggestions and high evaluation of our manuscript.
The authors successfully highlight the hydrophobic implications of baicalin on membrane lipid metabolism and carbohydrate metabolism. The authors do not clearly extrapolate the correlations between glucose transport at the membrane level and lipid metabolism modulating elements. The authors should try to suggest at least one potential mechanism involved in the successfully identified effects. For example, the existence of a baicalin structure that suggests a rigidity and a modulation of membrane fluidity, may suggest a modulation of the function of lipid shelves. This could justify the large number of metabolic modulations successfully identified by the authors.
In our review manuscript, we intended to include only fully verified information. Therefore, each piece of information included in the text is supported by the appropriate literature data. We completed some possible cellular mechanisms of baicalin effects on the basis of all the information about its antidiabetic action available in the literature. Figures summarizing the anti-diabetic effects of baicalin have been added. This indeed markedly raises the value of our manuscript. However, studies on cells is lacking, because it is impossible to explore diabetes in vitro. Moreover, some mechanisms of baicalin action are not fully explored and further research is needed. For example, Dai et al., (2018) have shown that baicalin is a competitive activator of CPT1A and thereby promotes FAs oxidation. However, these authors did not study the exact mechanism by which baicalin affects this enzyme.
Round 2
Reviewer 1 Report
Comments and Suggestions for Authors
Accept in present form
Reviewer 2 Report
Comments and Suggestions for Authors
The additional detailing of the potential mechanisms of action of baicalin increased the value of the article and the probability of obtaining citations.